# Development of an automatic segmentation system for anterolateral thigh flap perforators in maxillofacial reconstruction

Jisu Oh[1‡], Sungwon Ham[2‡], Jihye Heo[3], In-Seok Song[3*], Jee-Ho Lee[4*]

1 College of Medicine, University of Ulsan, Seoul, Republic of Korea, 2 Healthcare Readiness Institute for Unified Korea, Korean University Ansan Hospital, Ansan, Republic of Korea, 3 Department of Oral & Maxillofacial Surgery, Korea University Anam Hospital, Seoul, Republic of Korea, 4 Department of Oral and Maxillofacial Surgery, Asan Medical Center, College of Medicine, University of Ulsan, Seoul, Republic of Korea

‡ JO and SH are co first authors on this work.
* densis@korea.ac.kr (ISS); jeehoman@gmail.com (JHL)

## Abstract

The anterior thigh (ALT) flap is commonly used in reconstructive surgery, especially in maxillary reconstruction. Accurately identifying the perforator that supplies blood to the flap is critical for surgical success but is time-consuming and prone to variability since it is traditionally performed manually. Advances in artificial intelligence have shown convolutional neural networks (CNN) the potential to automate medical image segmentation. However, ALT flap perforator segmentation poses a unique challenge due to the small size of the perforator and its high anatomical variability. To address this challenge, we developed and validated a CNN-based automatic segmentation model for detecting ALT flap perforators on computed tomography angiography (CTA). Manual annotations of bilateral lateral femoral circumflex artery perforators were obtained from 80 patients using an image tracing program for comparison. The training for the development of an automatic segmentation system was then conducted based on these manual segmentation. The automatic segmentation system employed a two-stage cascaded approach: 2D detection with DeepLabv2 and 3D segmentation with ResNet152. Data augmentation techniques were applied to improve model generalization. Performance metrics included the dice similarity coefficient (DSC) and jaccard similarity coefficient (JSC). The automatic segmentation system achieved DSC and JSC values of 69.67 ± 1.48 and 67.81 ± 1.70, respectively. The distance differences between manual and automatic detection were 38.28 ± 15.52 mm on the left side and 31.96 ± 18.11 mm on the right side. The automatic segmentation system for ALT flap perforators demonstrates promising accuracy, highlighting its potential for clinical application. By reliably identifying perforator locations in CTA, the system can enhance the efficiency and precision of surgical planning, particularly for maxillofacial reconstruction.

**Data availability statement:** The data underlying this study are not publicly available due to the inclusion of potentially sensitive patient information, such as identifiable clinical imaging and medical records. Public sharing of these data would violate patient confidentiality and institutional privacy regulations. These restrictions have been imposed by the Institutional Review Board (IRB) of Asan Medical Center, which reviewed and approved the study protocol. Access to the data may be granted to qualified researchers upon reasonable request and subject to IRB approval. Requests for data access can be directed to the IRB of Asan Medical Center at: Email: irb@amc.seoul.kr Phone: +82-2-3010-7166 Address: 88, Olympic-ro 43-gil, Songpa-gu, Seoul, 05505, Republic of Korea.

**Funding:** This research was supported by a grant from the Korea Health Technology R&D Project through the Korea Health Industry Development Institute (KHIDI), funded by the Ministry of Health & Welfare, Republic of Korea (grant number: RS-2023-KH134676 to IS).

**Competing interests:** The authors have declared that no competing interests exist.

## Introduction

In 1984, Song et al. introduced the anterolateral thigh (ALT) flap as a septocutaneous vessel-based flap progressing between the rectus femoris and vastus lateralis muscles [1]. Since then, it has become a widely preferred option for maxillofacial reconstruction, known for its long pedicle length, suitable vessel diameter, adaptable design, and low donor site morbidity [2–4]. Based on these benefits, the ALT flap has become a workhorse flap for soft tissue reconstruction [5].

Previous studies with cadaver and image have shown significant variations of the ALT perforator anatomy in its origin, location, and course [6–8]. Up to 75% of ALT flap perforators originate from the descending branch of the lateral circumflex femoral artery, the largest branch of the profunda femoris, and up to 25% from the transverse branch [2,5,8]. These perforators can also be found on the ascending and lateral branches of the lateral circumflex femoral artery [9]. The progression of perforators can be categorized as septocutaneous or musculocutaneous; in the previous study, more than 85% of cases exhibited a musculocutaneous course [2,6,9,10]. The anatomical variability of ALT flap perforators poses challenges in preoperative planning, including the flap design.

Color Doppler ultrasound (CDU) and computed tomography angiography (CTA) are commonly used for preoperative perforator detection. CDU is portable, provides information on vascular flow and tissue, and allows intra- and postoperative monitoring [11,12]. However, CDU has a steep learning curve and suffers from interobserver variability [12,13]. CTA provides greater sensitivity and specificity, aiding in accurate flap design, but is time-intensive, reliant on image quality, and can miss intramuscular perforators due to muscle radiodensity [14]. The subjectivity of perforator segmentation can lead to discrepancies between preoperative plans and surgical findings, often requiring intraoperative adjustments [15,16].

To address these limitations, deep learning and automatic detection have been applied in other medical fields, such as assessing diabetic retinopathy and lung nodule detection [17,18]. Studies have also explored automatic segmentation for preoperative perforator detection in breast reconstruction. However, there is currently no research on automatic segmentation for ALT flap perforators.

[15,16] Preoperative image detection of ALT flap perforators with CTA is widely used to aid ALT flap design. However, anatomical variability in ALT perforators can complicate intraoperative procedures, leading to extended flap harvest times, potential flap design alterations, and even a need to switch to the contralateral leg, impacting donor site morbidity. Studies have shown that 3% to 5% of patients lack suitable perforators, requiring alternative flap design [7,19]. Therefore, increasing preoperative detection accuracy is crucial for improved surgical outcomes.

This study aims to develop an automatic segmentation system for detecting ALT flap perforators, a key component in maxillofacial reconstruction. Ground truth data were generated through manual segmentation of CTA images by an experienced clinician, and a deep learning model was trained using both 2D and 3D image inputs. Retrospective validation was conducted on additional datasets to assess accuracy in perforator localization and clinical applicability.

## Methods

The study protocol was approved by the Institutional Review Board of Asan Medical Center (IRB No. 2023−1372). Data used for this study were accessed for research purposes between 30 March 2023 and 31 Aug 2023.

### Study subjects and data collection

This study enrolled 80 patients who underwent preoperative CTA for maxillofacial reconstruction at the Department of Oral and Maxillofacial Surgery, Asan Medical Center between 1 March 2021 and 30 June 2023. The exclusion criteria were (1) patients with missing CTA data (n = 4), (2) patients in whom the collected CTA did not run on the mapping software (n = 5), and (3) patients who had poor quality of CTA with artifacts (n = 2). The mean age was 62.1 years (range, 35–84 years), and the mean BMI was 23.5 kg/m$^2$ (range, 18.99–34.85 kg/m$^2$).

### Manual segmentation of ALT flap perforator

ALT flap perforators were manually segmented using AVIEW Modeler (version 1.1.42.7, Coreline soft, Seoul, Korea). After uploading anonymized CTAs to the software, the 3-mm cut of the CTA was converted to 1-mm cut using an image conversion option to accurately predict the ALT flap perforator. The region of interest (ROI) was manually defined in the coronal plane, extending from the iliac crest to the superior margin of the patella. Within this ROI, the lateral circumflex femoral artery perforator was manually segmented in a continuous manner, starting from the femoral artery near the hip joint to its terminal point on the thigh skin (Fig 1). The segmentation was performed slice-by-slice based on visual inspection by a clinician, without the use of automated techniques such as seed-growing or centerline tracking. These manual annotations served as the ground truth for the subsequent training of the automatic segmentation model.

The progression of the perforator was categorized as septocutaneous or musculocutaneous (Fig 2). A total of 69 patients were included for analysis. Among these, 56 patients were assigned to the training and validation set, and 13 patients to the test set. Each of the CTA scans from 56 patients was divided into left and right halves, thereby eliminating the distinction between sides. This approach effectively doubled our dataset, which led to the creation of 112 usable patients for a more comprehensive analysis. Then, the expanded dataset was strategically divided into three subsets, training, validation, and testing, at a ratio of 8:1:1. Subsequently, to further enhance the robustness of our model testing, additional 13 scans were included in the test set.

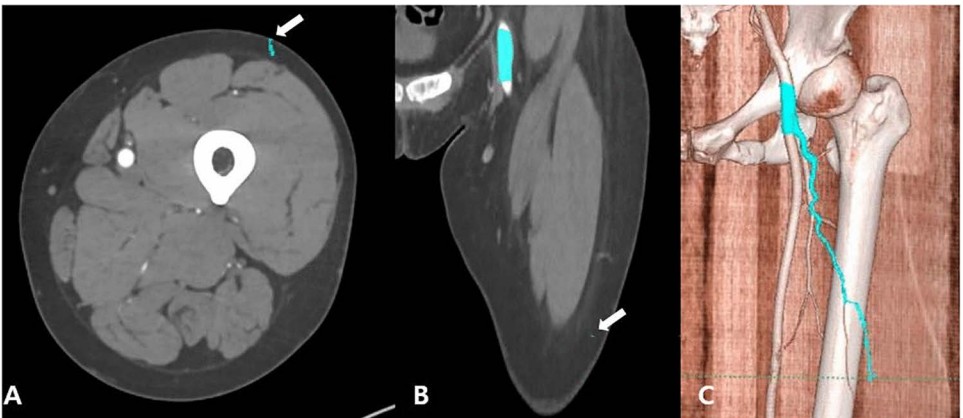

**Fig 1. Axial (A), coronal (B), and 3D (C) views of computed tomography angiography.** Manual segmentation from the femoral artery near the hip joint to the lateral circumflex femoral artery perforator was performed continuously to the terminal point reaching the thigh skin.

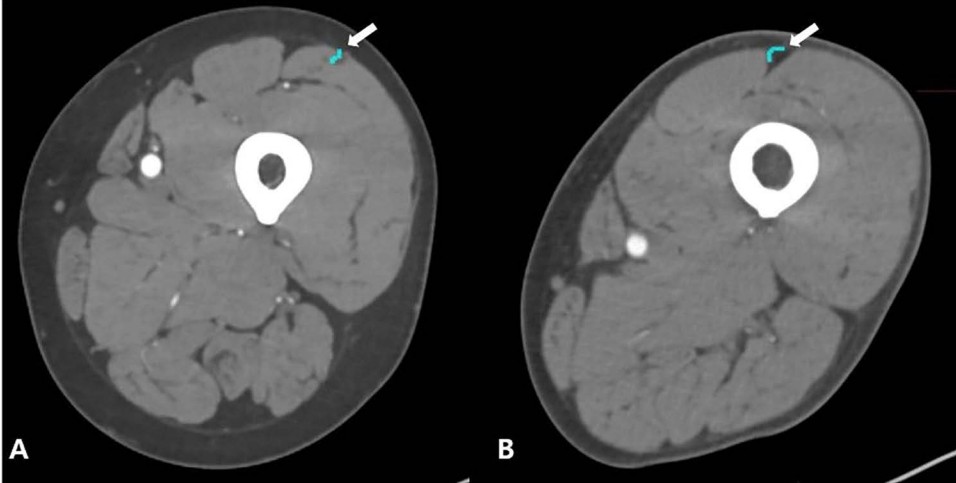

**Fig 2. The axial views show the perforator progression as musculocutaneous (A) and septocutaneous (B).**

## Network architectures

This study employed a cascaded approach for the detection and segmentation of the perforators [20]. The network architectures were developed in two stages (Fig 3). The first stage of the process involved the use of DeepLabv2, a convolutional neural network known for its efficiency in semantic image segmentation [21]. Complete 2D CTA images were introduced into this network, and their original resolution of 512×512 pixels was maintained to preserve details important

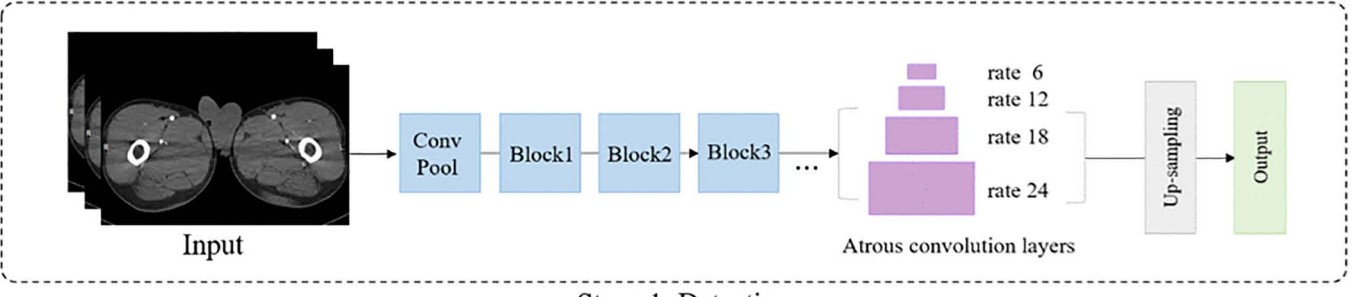

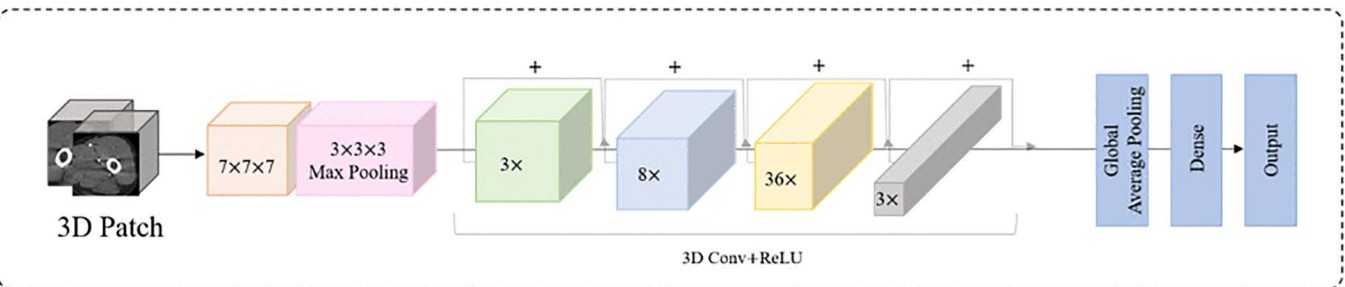

**Fig 3. Network architectures consisting of two stages.**

for accurate detection. The learning rate was meticulously calibrated to 0.0001 to achieve a balanced compromise between rapid convergence and avoidance of the risk of overshooting the minimum during training. After the assessment, a batch size of 16 was selected and the DeepLabv2 model was subjected to extensive training, spanning 500 trials to fine-tune its performance.

After completing the 2D detection phase, the second stage involving 3D segmentation was initiated. In this stage, 3D patches centered on the previously detected Perforator regions were extracted. These patches were 64×64×64 voxels in size; this size was selected as it was computationally manageable in the 3D ResNet152 model while still sufficiently showing the perforation regions in detail for effective segmentation. The patches were then entered into the 3D ResNet152 model for volume data learning, and network-wide rectified linear unit activation was used, known as the efficiency of deep learning models [22]. The model learning rate was set to 0.001, with a batch size of 32 chosen for efficient 3D processing. Furthermore, Various data augmentation techniques were implemented to enhance the model's generalization ability. The scans were randomly rotated up to 20° to mimic different anatomical orientations. Image sizes were adjusted between 90% and 110% of their original dimensions to account for anatomical size variations. Additionally, random horizontal and vertical translations were applied to simulate positional differences, and brightness and contrast adjustments were made to reflect the variability seen in clinical images.

## Evaluation criteria

**Primary Evaluation (Training Performance).** Two statistical metrics were mainly used in the evaluation of the performance of our segmentation model: Dice Similarity Coefficient (DSC) and Jaccard Similarity Coefficient (JSC) [23]. The DSC, a measure ranging from 0 to 1, quantifies the overlap between two samples, where 1 denotes perfect overlap and 0 no overlap at all. It is calculated as follows:

$$\frac{2 \times |A \cap B|}{|A| + |B|}$$

where A and B denote the actual and predicted regions, respectively. The JSC also evaluates similarity with values between 0 and 1, but it measures the proportion of overlap to the total size of the combined samples. It is calculated as follows:

$$\frac{|A \cap B|}{|A \cup B|}$$

All accuracy metrics were calculated per patient by averaging across the segmented slices of each case. Although the segmentation was performed at the slice level, the final evaluation was based on patient-level aggregation to reflect clinical relevance.

## Secondary Evaluation

***Clinical evaluation.*** To assess clinical applicability, additional manual segmentation was performed by the same clinician for 13 test patients following automatic segmentation. The results of both manual and automatic segmentations were uploaded to the same dataset and visualized on 3D CTA images. The image threshold was adjusted so that the terminal point of the perforator appeared as a distinct point on the skin surface (Fig 4). The Euclidean distance between the manually and automatically detected terminal points was calculated. The secondary outcome measure was the spatial localization error, defined as the Euclidean distance between manually and automatically detected terminal skin points of each perforator. Additionally, based on the location of the manually annotated perforator, the corresponding automatic detection was classified into four quadrants: mediosuperior, medioinferior, distosuperior, and distoinferior.

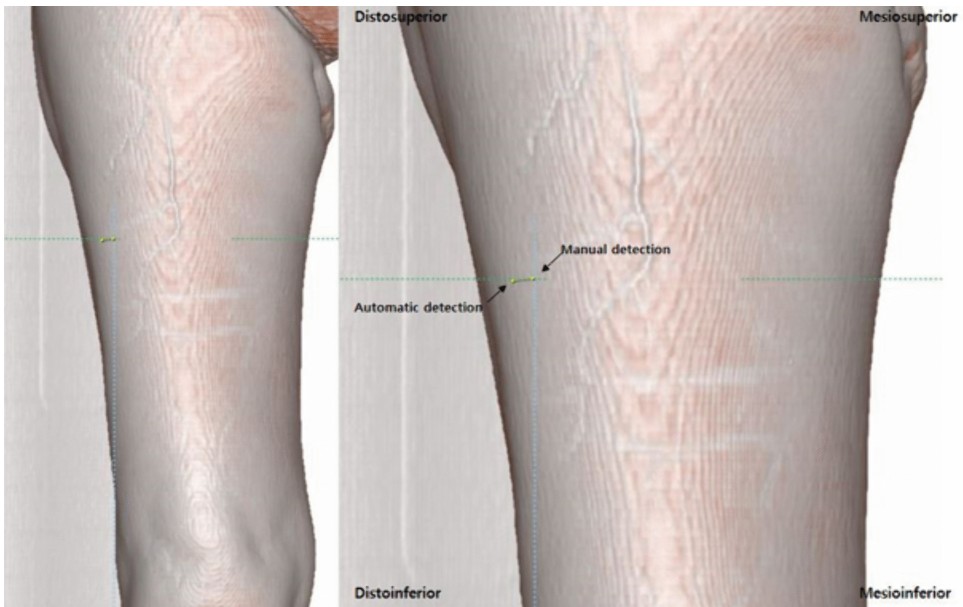

**Fig 4. After superimposing manually and automatically segmented perforators on the thigh skin, the difference in the coordinate values between two points was calculated.**

**Subgroup Analysis.** To explore anatomical and demographic influences on segmentation performance, subgroup analyses were performed. Mann–Whitney U tests were used to compare DSC and JSC values between groups stratified by gender, laterality (left vs. right thigh), and perforator progression type (septocutaneous vs. musculocutaneous). The Kruskal–Wallis test was used to compare segmentation performance across BMI categories. BMI was classified according to World Health Organization (WHO) cutoffs: underweight ($<18.5\,kg/m^2$), healthy weight ($18.5–24.99\,kg/m^2$), overweight ($25–29.99\,kg/m^2$), and obese ($\geq30\,kg/m^2$) [24]. Because DSC and JSC values did not meet normality assumptions, non-parametric tests were used for all group comparisons, and Mann–Whitney U or Kruskal–Wallis tests were selected accordingly.

### Statistical analysis

All statistical analyses were performed using SPSS version 22.0 (IBM, Chicago, IL, USA) with a significance level of $p < 0.05$.

## Results

### Subject distribution

The datasets for the gold standard consisted of 26 men and 30 women (average age, $68.1 \pm 9.6$ years) for training and validation. The datasets for the test included 10 men and 3 women (average age, $61.38 \pm 13.07$ years). The progression type of the perforator was determined according to sex. In the training dataset, 74.1% of septocutaneous and 25.9% of musculocutaneous perforators were investigated in men and 41.4% of septocutaneous and 58.6% of musculocutaneous perforators in women (Table 1).

### Training evaluation

In this study, the ALT flap perforator starting from the femoral artery near the hip joint, passing through the lateral circumflex femoral artery and reaching the thigh skin, was detected via automatic segmentation (Fig 5). The validity of automatic

**Table 1. Progression type of perforators according to gender.**

| N (%) | | Septocutaneous | | Musculocutaneous | | |
|---|---|---|---|---|---|---|
| | | Lt. | Rt. | Lt. | Rt. | Total |
| Training | Male | 21(38.9) | 19(35.2) | 6(11.1) | 8(14.8) | 54 |
| | Female | 11(19.0) | 13(22.4) | 18(31.0) | 16(27.6) | 58 |
| | Total | 32(28.6) | 32(28.6) | 24(21.4) | 24(21.4) | 112 |
| Validation | Male | 6(30.0) | 6(30.0) | 4(20.0) | 4(20.0) | 20 |
| | Female | 1(16.7) | 1(16.7) | 2(33.3) | 2(33.3) | 6 |
| | Total | 7(26.9) | 7(26.9) | 6(23.1) | 6(23.1) | 26 |

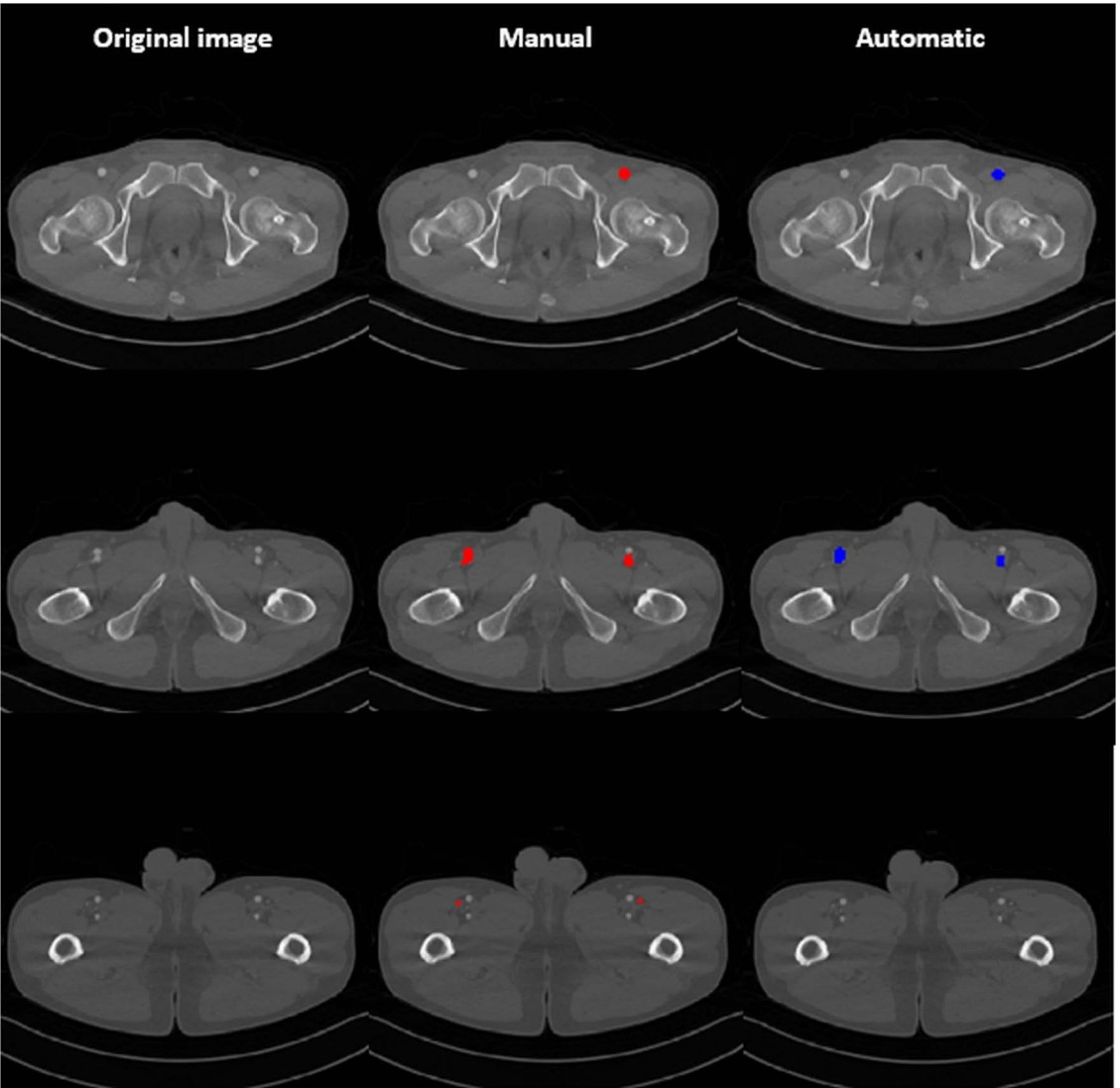

**Fig 5. Axial images of original, manual, and automatic detections.** The results of the manual and automatic detections showed similarity. As the diameter of arteries got smaller, discontinued portions were confirmed.

segmentation for the lateral circumflex femoral artery perforator was examined using DeepLabv2 on 2D CTA and 3D Res-Net152 on 3D CTA. The average DSC and JSC values were 69.67±1.48 and 67.81±1.70 for the training and validation datasets and 69.34±0.83 and 67.47±0.96 for the test datasets, respectively (Table 2).

When analyzing the DSC and JSC values according to the left and right sides of the thigh, the DSC values were 69.73±1.64 for the left side and 67.82±1.58 for the right side ($p=0.005$) and the JSC values were 68.95±1.69 for the left side and 67.14±1.89 for the right side ($p=0.026$), which showed a higher degree of accuracy with the automatic segmentation on the left side. On the other hand, there were no statistical differences in DSC and JSC values according to gender, perforator progression type, and BMI (Table 3).

## Clinical evaluation

In manual and automatic detection, 2 points of perforators reaching to the thigh skin were superimposed and the position difference was calculated. The average difference on the right side was 31.96±18.11 mm, whereas the average difference on the left side was 38.28±15.52 mm. Although these values represent several centimeters of deviation, they reflect the inherent difficulty of localizing small-caliber perforators on CTA and still provide clinically meaningful guidance for preoperative flap planning.

**Table 2. Dice similarity coefficient and jaccard similarity coefficient values of automatic segmentation for lateral circumflex femoral artery perforators.**

| Mean (%) ± SD | Training and validation | Test |
|---|---|---|
| DSC | 69.67±1.48 | 69.34±0.83 |
| JSC | 67.81±1.70 | 67.47±0.96 |

SD: standard **deviation**, DSC: dice similarity coefficient, JSC: jaccard similarity coefficient

**Table 3. Comparison of DSC and JSC values according to gender, left and right thighs, perforator progression type, and BMI.**

| Mean (SD) | | DSC | JSC |
|---|---|---|---|
| Gender† | | | |
| | Female | 68.78(1.18) | 66.72(0.67) |
| | Male | 69.51(0.69) | 67.70(0.94) |
| | *p* value | 0.287 | 0.112 |
| Location† | | | |
| | Left | 69.73(1.64) | 68.95(1.69) |
| | Right | 67.82(1.58) | 67.14(1.89) |
| | *p* value | 0.005* | 0.026* |
| Perforator progression† | | | |
| | Septocutaneous | 69.33(1.58) | 67.18(1.93) |
| | Musculocutaneous | 69.36(1.85) | 67.84(1.49) |
| | *p* value | 0.860 | 0.297 |
| BMI‡ | | | |
| | Healthy weight (18.5–24.99 kg/m2) | 69.27(0.91) | 67.31(0.99) |
| | Overweight (25–29.99 kg/m2) | 69.82(0.58) | 68.19(0.90) |
| | Obese (≥30 kg)/m2) | 69.04(0.00) | 67.60(0.00) |
| | *p* value | 0.459 | 0.392 |

*: statistically significant with $p<0.05$, †: Mann-Whitney u-test, ‡: Kruskal-Wallis test.

SD: standard deviation, DSC: dice similarity coefficient, JSC: jaccard similarity coefficient, BMI: body mass index.

## Discussion

This study presents an automatic segmentation system for ALT flap perforators based on convolutional neural networks, addressing the limitations of manual CTA-based detection, which can be time-consuming and highly operator-dependent. To evaluate performance, the system's accuracy was measured using mIoU, DSC, and JSC. DSC and JSC scores ranged from 0.750 to 0.7065 and 0.6614 to 0.6882, respectively, levels that may be considered clinically acceptable given the fine structure of the target vessels [25,26]. Because the dataset size was relatively modest, expanding the cohort, particularly through multi-center integration, is expected to further improve model robustness and reduce performance variability.

In manual flap perforator segmentation, vascular courses were classified as musculocutaneous or septocutaneous. Studies report 17% to 85% of cases with musculocutaneous vascular courses [2,3,6,9,10]. When perforators follow a musculocutaneous route through the vastus lateralis, flap harvest times increase, leading to higher donor site morbidity [2,4,5]. This study found 56.5% of patients had septocutaneous perforators, while 43.5% had musculocutaneous. The septocutaneous course was more common in men (70.3%), while the musculocutaneous course was prevalent in women (59.4%). Due to high radiodensity in muscular areas, musculocutaneous perforator detection appears operator-dependent. Although the segmentation model was not designed to differentiate between septocutaneous and musculocutaneous perforators, it demonstrated similar performance across both types. Subgroup analysis showed no statistically significant difference in DSC or JSC, suggesting that the model was robust to variations in perforator course. Nevertheless, the most challenging cases often involved musculocutaneous perforators that traveled through the muscle before reaching the subcutaneous layer, which may still affect detection accuracy in certain individuals. To assess annotation reliability, an additional inter-rater validation was performed. A second oral and maxillofacial surgery resident independently annotated 10 randomly selected cases, and agreement between the two annotators was quantified using the Dice similarity coefficient, demonstrating high consistency (mean DSC = $0.88 \pm 0.04$).

Each subject had varying physical and demographic characteristics, such as age, sex, muscular density, and thickness of the subcutaneous tissue. This may affect the detection accuracy and the time required for manual segmentation. A previous study reported that there was no difference in the number of perforators between the dominant and nondominant legs during the ALT flap perforator detection and that it was not affected by the thickness of the subcutaneous tissue and muscle [27]. When statistical analysis was performed on the test datasets, the accuracy of the left thigh between the manual and automatic segmentation was significantly higher than that of the right thigh. On the other hand, there was no statistical difference in accuracy depending on the type of perforator progression (DSC: $p=0.860$), gender (DSC: $p=0.287$), or BMI (DSC: $p=0.459$), which is considered related to subcutaneous tissue thickness. This may be attributed to the model's ability to extract consistent vascular features regardless of anatomical variations. Moreover, the limited sample size and standardized CTA acquisition protocol may have reduced intersubject variability, potentially masking subtle differences across subgroups.

In this study, segmentation was adopted to localize the perforator boundaries precisely, as exact positional information was required for clinical applicability [28–30].

Our study initially conducted deep learning using DeepLabv2 with 2D images. In fact, the mIoU values for the analysis using 2D images were slightly higher than those for the analysis using 3D images. However, in the process of converting CTA from 3- to 1-mm cuts, the number of slices on average increased to more than 1000, and the image segmentation loss was partially detected. As false positives were likely to be increased in these circumstances, evaluation of overall connectivity using 3D ResNet152 with 3D images was attempted. A previous study demonstrated that 2D image segmentation was more effective than 3D image segmentation using abdominal CT images [31]. Other studies conducted segmentation using 2D and 3D images interactively. Similarly, our study used 2D and 3D images in a cascaded manner to compensate for the strengths and weaknesses of each image [32–34].

The main trunk starting from the femoral artery near the hip joint had a large diameter, showing high agreement with manual and automatic detections, but as it went toward the terminal branches reaching the skin, a relatively disconnected

area occurred in the middle area of the artery compared with the ground truth. However, because detection of perforators terminating at the thigh skin is crucial in actual intraoperative procedures, the discontinuity of the middle parts of the artery may not be a critical problem in the clinical application of the automatic perforator segmentation system. When the region of interest (ROI) was divided into superior, middle, and inferior thirds, segmentation performance was highest in the superior third (DSC: 72.1±1.8; JSC: 70.4±2.1), followed by the middle (DSC: 68.3±1.7; JSC: 67.0±1.9) and inferior thirds (DSC: 65.7±2.0; JSC: 64.2±2.4). This trend may reflect anatomical features, such as the larger diameter of the main arterial trunk in the proximal region near the femoral origin.

To provide clinical context, we refer to the ABC system, which was designed to detect dominant ALT flap perforators based on consistent vascular patterns observed in the lower limbs of Western populations. In this previous study, 89% of reliable perforators were detected 3 cm lateral to the midpoint of the line between the anterior superior iliac spine and the superior lateral corner of the patella, referred to as the A-P line [35]. When test datasets were retrospectively evaluated, all 26 ALT flap perforators were located lateral to the A-P line.

For clinical evaluation, the distance between the manually and automatically detected skin termination points of the perforator was calculated. Although the mean deviation was relatively large for both thighs (38.28±15.52 mm on the left, 31.96±18.11 mm on the right), no statistically significant difference was observed. These findings suggest that while the model demonstrates robust detection capability, variability remains in pinpoint accuracy, which may be influenced by factors such as gender, age, or perforator course. Further research with a larger dataset and refined labeling strategies may improve spatial precision and enhance clinical applicability in flap design. Although intraoperative validation was not performed in this study, the evaluation method—comparing AI-predicted perforator locations with manual annotations on CTA—reflects a clinically feasible approach that aligns with preoperative planning practices. Previous studies have shown that preoperative CTA mapping significantly improves surgical outcomes by enabling precise identification of reliable perforators and optimizing flap design [36]. Therefore, assessing the concordance between manual and automatic detection in 3D space offers clinically meaningful insight into the usability of the proposed model. However, the localization error in this study was calculated solely as the Euclidean distance between terminal skin points without reference to anatomical surface landmarks such as the ASIS or patella. Incorporating landmark-normalized measurements in future work may enhance clinical applicability by providing a more precise correspondence to intraoperative surface anatomy. Moreover, ultrasound provides real-time identification of perforators at the skin surface, and future comparative work integrating CTA with ultrasound-based localization may further improve preoperative accuracy.

Interestingly, segmentation performance showed a statistically significant difference between the left and right thighs, with higher DSC and JSC values observed on the left side (DSC: 69.73 ± 1.64 vs. 67.82 ± 1.58, $p$ = 0.005; JSC: 68.95 ± 1.69 vs. 67.14 ± 1.89, $p$ = 0.026). While the precise cause remains unclear, this discrepancy may be attributed to factors such as anatomical asymmetry, differences in image acquisition angle, or leg dominance in muscle and vascular development. Further investigation using a larger and more balanced dataset is warranted to determine whether this finding reflects a consistent pattern or an incidental variation.

This study has several limitations. The sample size used to train the deep learning model was relatively small, and external validation using additional datasets is needed to further improve the accuracy of the automatic segmentation system. As aforementioned, the detection accuracy for perforators on the skin was slightly lower, so retraining is necessary by reducing the ROI. The ROI set in our study was from the anterior superior iliac crest to directly superior to the patella. It would be helpful to conduct additional study by reducing the ROI based on the ABC system. As manual detection is time-consuming, it seems necessary to evaluate efficiency by comparing the difference in the time required for manual and automatic segmentations. Although the manual annotations distinguished septocutaneous and musculocutaneous perforators, the current model did not incorporate this classification, which remains an area for future methodological extension.

Additionally, segmentation performance tended to decrease toward the distal ends of the perforators, likely due to the progressive reduction in vessel diameter. While the main trunk arising from the femoral artery is relatively easy to identify, clear separation of the mid-branch and distal perforator segments is challenging even for experienced clinicians due to

their small caliber and variable course on CTA. Direct evaluation across perforator calibers was not feasible because the spatial resolution of CTA is insufficient to reliably measure diameter differences in distal perforator branches. This limitation should be addressed in future work using higher-resolution imaging or intraoperative validation. In addition, although the manual annotations distinguished septocutaneous and musculocutaneous perforators, the current model did not incorporate this classification. Integrating perforator-type differentiation into future models would enhance the clinical relevance of automated preoperative planning. Although subgroup analysis of DSC and JSC by anatomical thirds (proximal, middle, distal) was considered, the difficulty in consistently defining these regions in clinical imaging limited its feasibility. In a further study, we plan to stratify segmentation performance according to vessel diameter or anatomical subregions, which may help overcome the current limitations, particularly in the distal segments.

## Conclusions

In conclusion, this study demonstrates the feasibility of deep learning–based automatic segmentation for detecting ALT flap perforators using CTA. While the model showed clinically acceptable performance, particularly in terms of segmentation overlap, certain limitations such as spatial accuracy in terminal point localization remain. Further research with expanded datasets and optimized training strategies is needed to improve precision and support broader clinical adoption.

## Author contributions

**Conceptualization:** In-Seok Song.

**Data curation:** Jee-Ho Lee.

**Methodology:** Sungwon Ham.

**Project administration:** In-Seok Song.

**Resources:** Jihye Heo.

**Software:** Sungwon Ham.

**Validation:** Jisu Oh.

**Writing – original draft:** Sungwon Ham, Jisu Oh.

**Writing – review & editing:** Jee-Ho Lee.

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
