## [Decision Letter · Decision Letter 0]

17 Mar 2025

Dear Dr. Song,

Thank you for submitting your manuscript to PLOS ONE. After careful consideration, we feel that it has merit but does not fully meet PLOS ONE’s publication criteria as it currently stands. Therefore, we invite you to submit a revised version of the manuscript that addresses the points raised during the review process.

We look forward to receiving your revised manuscript.

Kind regards,

Shimpei Miyamoto

Academic Editor

PLOS ONE

**Journal Requirements:**

1. When submitting your revision, we need you to address these additional requirements. Please ensure that your manuscript meets PLOS ONE's style requirements, including those for file naming. The PLOS ONE style templates can be found at https://journals.plos.org/plosone/s/file?id=wjVg/PLOSOne_formatting_sample_main_body.pdf and https://journals.plos.org/plosone/s/file?id=ba62/PLOSOne_formatting_sample_title_authors_affiliations.pdf 2. Please note that PLOS ONE has specific guidelines on code sharing for submissions in which author-generated code underpins the findings in the manuscript. In these cases, we expect all author-generated code to be made available without restrictions upon publication of the work. Please review our guidelines at https://journals.plos.org/plosone/s/materials-and-software-sharing#loc-sharing-code and ensure that your code is shared in a way that follows best practice and facilitates reproducibility and reuse. 3. Thank you for stating the following in the Acknowledgments Section of your manuscript: This research was supported by a grant of the Korea Health Technology R&D Project through the Korea Health Industry Development Institute (KHIDI), funded by the Ministry of Health & Welfare, Republic of Korea (grant number: HI23C0162). We note that you have provided funding information that is not currently declared in your Funding Statement. However, funding information should not appear in the Acknowledgments section or other areas of your manuscript. We will only publish funding information present in the Funding Statement section of the online submission form. Please remove any funding-related text from the manuscript and let us know how you would like to update your Funding Statement. Currently, your Funding Statement reads as follows:  The author(s) received no specific funding for this work. Please include your amended statements within your cover letter; we will change the online submission form on your behalf. 4. We note that you have indicated that there are restrictions to data sharing for this study. For studies involving human research participant data or other sensitive data, we encourage authors to share de-identified or anonymized data. However, when data cannot be publicly shared for ethical reasons, we allow authors to make their data sets available upon request. For information on unacceptable data access restrictions, please see http://journals.plos.org/plosone/s/data-availability#loc-unacceptable-data-access-restrictions.  Before we proceed with your manuscript, please address the following prompts: a) If there are ethical or legal restrictions on sharing a de-identified data set, please explain them in detail (e.g., data contain potentially identifying or sensitive patient information, data are owned by a third-party organization, etc.) and who has imposed them (e.g., a Research Ethics Committee or Institutional Review Board, etc.). Please also provide contact information for a data access committee, ethics committee, or other institutional body to which data requests may be sent. b) If there are no restrictions, please upload the minimal anonymized data set necessary to replicate your study findings to a stable, public repository and provide us with the relevant URLs, DOIs, or accession numbers. Please see http://www.bmj.com/content/340/bmj.c181.long for guidelines on how to de-identify and prepare clinical data for publication. For a list of recommended repositories, please see https://journals.plos.org/plosone/s/recommended-repositories. You also have the option of uploading the data as Supporting Information files, but we would recommend depositing data directly to a data repository if possible. Please update your Data Availability statement in the submission form accordingly.

Reviewers' comments:

Reviewer's Responses to Questions

**Comments to the Author**

1. Is the manuscript technically sound, and do the data support the conclusions?

Reviewer #1: Partly

Reviewer #2: Partly

Reviewer #3: Yes

2. Has the statistical analysis been performed appropriately and rigorously?

Reviewer #1: Yes

Reviewer #2: Yes

Reviewer #3: N/A

3. Have the authors made all data underlying the findings in their manuscript fully available?

Reviewer #1: Yes

Reviewer #2: No

Reviewer #3: Yes

4. Is the manuscript presented in an intelligible fashion and written in standard English?

Reviewer #1: Yes

Reviewer #2: Yes

Reviewer #3: No

**Reviewer #1:** I have several points for this paper to be finalized.

Abstract

1. The purpose of this study in abstract is written twice.

M&M

2. Study subjects and data collection - all reviewed patient's information should be presented in this section (gender, age, BMI)

3. training, validation and test set number is described in very confused way. In total, manual segmentation for 69 patients (138 image sets) were done?

4. The term "ground truth and manual mapping" is used in confusion way. For train as well as test sets, ground truth is just ground truth. Also, the term mapping cannot be used as the replacement of segmentation.

5. The manual segmentation method should be described with more detail. Did clinician draw boundary of the vessel to determine ROI? Or identified center and used seed grown method? This is also related to the clinical evaluation.

6. In clinical evaluation, the difference between the center of ROI was obtained? If it is so, the determining method of the center of ROI should be stated. It is not clearly written.

7. It is difficult to understand how the method implemented in the clinical evaluation can be considered a clinical evaluation. It would be better to refer to the previous literature further. This evaluation method dose not fully understand as the way that clinician used in actual clinical conditon.

Result

Overall result is very confusing because it shows too much information. Some are not related to the study purpose and some are not necessary to support study hypothesis.

8. Subject distribution - I would suggest to present a table for patient's general information. Also, is it necessary to to describe the type of perforator according to sex?

9. There must be explanation for the difference between the right and left side.

10. Some results are not suitable for the study purpose, for example figure 2. This study is not for showing distribution of vessel position.

Discussion

11. the first paragraph should be in introduction section.

12. The third paragraph is not very important according to the study purpose, except the last sentence. Although, the last sentence seems broken and don't understand. (This study's most challenging detections involved perforators traveling through muscles before reaching subcutaneous layers.)

13.In 4th paragraph, "the progression course of the artery could be confirmed more clearly in younger women with sufficient subcutaneous tissue thickness in.." -> This part was not stated either in method nor result. It cannot be discussed as the readers do not have any idea about it.

14.In 4th paragraph, "On the other hand, there was no statistical difference in accuracy depending on the type..." This part needs to be discussed about.

15. 5th paragraph can be removed or reduced.

16. 7th paraphragh, "The main trunk starting from the femoral artery near the hip joint had a large diameter...." I cannot find any description about this information, it suddenly appeared in disscusion only. I would suggest to present result of DSC, JSC if there is significancy according to the superior1/3, middle1/3, inferior1/3 part. If it is presented, the above sentence can be supported.

17. 8th paragraph is not very related to the study purpose or result?

18. 9th paraphraph is the repeatition of the previous paragraphs, needs to be revised.

19. 10th paragraph, "First, the n value to proceed": "n value" -> sample size

**Reviewer #2:** Authors of this study propose an innovative methodological solution to facilitate the identification of the vessels of the ALT pedicle.

The study focuses on analysis and modeling, by a neural network of the path, of the perforating artery, and its comparison with manual modeling by an experimented operator.

The objective of the study seems to be the feasibility of modeling the path of pedicle by AI software. The comparison with manual modeling serves as extrinsic validation.

The primary endpoint is the accuracy of the location of the perforator on imaging. The means of evaluation is the comparison of projection concordance tests on imaging. The secondary endpoints seem to be the influence of gender, side, BMI and muscular or septal location of the perforators.

The study has a small number of patients: 56 patients for machine learning (plus 11 excluded), 13 patients for the control after learning. Both legs were analyzed for each patient. The software performed the detection on the 2D scanner, then reconstructed in 3D the path of the vessel, and the projection of the perforator to the skin. Changes in size, contrast, and angulation of the images were applied randomly to eliminate algorithmic bias. Concordance tests for the detection of the perforator on the slices were performed.

The results found a statistically significant projection concordance for the left side, not significant for the right side. The discussion clarifies some aspects and puts this study into perspective with the existing literature. In addition, they take into consideration the limitations of this study, particularly in terms of patient numbers.

This paper is interesting, providing a relevant solution to simplify the identification of perforators, however, it seems to me that the work is not entirely complete.

The objective of the study seems a little imprecise. Is the objective to demonstrate the feasibility of automatic segmentation, or to show that automatic segmentation performs as well as manual segmentation?

The primary judgment criterion could be specified: is it a comparison on 2D slices or on 3D reconstructions?

It seems relevant to me to formalize the expression of the secondary judgment criteria, these do not seem to be perfectly defined in the article. The comparison of projection concordance tests seems to be adapted to the demonstration of the objective of the study and perfectly conducted. The statistical tests seem relevant to me.

In the results and the discussion, I did not understand whether the computer analysis allowed to detect the septal or muscular path of the perforators, or whether the margin of error did not allow to conclude.

The conclusion of the study could be specified. We can conclude that automatic segmentation is possible, but, within the limits of this experimental procedure, seems to still lack a little precision.

**Reviewer #3:** This study addresses the challenge of identifying perforators in the anterolateral thigh (ALT) flap for maxillofacial reconstruction by developing an automatic segmentation system, directly serving clinical practice and aligning with the practical needs of plastic surgery. Additionally, it pioneers the application of deep learning in the automatic segmentation of ALT flap perforators, filling a gap in this field and demonstrating significant innovation. However, some issues may need to be addressed. 1. The small sample size could limit the generalizability of the model, especially in clinical settings where diverse anatomical variations exist. Although the authors use manual mapping as the ground truth for training the model, relying on a single researcher for this step introduces a potential source of bias. 2. The reported distance differences between manual and automatic detection of perforators (e.g., 38.28 ± 15.52 mm on the left side) suggest some degree of inaccuracy. This could potentially be problematic for clinical applications where precise identification of perforators is crucial for successful surgical planning. 3. The references are outdated and may not reflect the latest advancements in imaging technologies and the application of deep learning in medical practices.

**Do you want your identity to be public for this peer review?** For information about this choice, including consent withdrawal, please see our Privacy Policy

Reviewer #1: No

Reviewer #2: **Yes:** Franck Delanoë

Reviewer #3: No

---

## [Author Response · Author response to Decision Letter 1]

4 Aug 2025

We sincerely appreciate the valuable feedback provided by the reviewers and editor. We have addressed all reviewer and editor comments in detail in the uploaded “Response to Reviewers” document.

---

## [Decision Letter · Decision Letter 1]

13 Oct 2025

Dear Dr. Song,

Thank you for submitting your manuscript to PLOS ONE. After careful consideration, we feel that it has merit but does not fully meet PLOS ONE’s publication criteria as it currently stands. Therefore, we invite you to submit a revised version of the manuscript that addresses the points raised during the review process.

We look forward to receiving your revised manuscript.

Kind regards,

John Minh Le, MD, DDS

Academic Editor

PLOS ONE

Journal Requirements:

Reviewers' comments:

Reviewer's Responses to Questions

**Comments to the Author**

Reviewer #2: All comments have been addressed

Reviewer #3: All comments have been addressed

Reviewer #4: (No Response)

2. Is the manuscript technically sound, and do the data support the conclusions?

Reviewer #2: Yes

Reviewer #3: Yes

Reviewer #4: Yes

3. Has the statistical analysis been performed appropriately and rigorously?

Reviewer #2: Yes

Reviewer #3: Yes

Reviewer #4: Yes

4. Have the authors made all data underlying the findings in their manuscript fully available?

Reviewer #2: Yes

Reviewer #3: Yes

Reviewer #4: No

5. Is the manuscript presented in an intelligible fashion and written in standard English?

Reviewer #2: Yes

Reviewer #3: Yes

Reviewer #4: Yes

Reviewer #2: Thank you for the corrections you have made; they make your manuscript much more relevant.

In essence, it does indeed seem necessary to increase the number of subjects in the machine learning process to improve accuracy. The margin of error is a little too large for clinical application, and the lack of differentiation between the septal and muscular pathways is regrettable.

As a supplementary conclusion to this work, it could be suggested that this methodology may be applied to other perforator flaps.

Thank you for this work.

Reviewer #3: This is an excellent research. The author's response cleared up my doubts. I have no further comments!

Reviewer #4: This manuscript presents a deep learning–based approach for automatic segmentation of anterolateral thigh (ALT) flap perforators on computed tomography angiography (CTA). The clinical relevance is clear: accurate perforator identification is critical for flap harvest in maxillofacial reconstruction, yet manual mapping is time-consuming and variable. The novelty lies in applying convolutional neural networks (CNNs) to this niche domain, using a cascaded 2D/3D segmentation strategy. Overall, the study is innovative, clinically relevant, and well-structured. However, several methodological and interpretive aspects merit clarification and strengthening.

My major Concerns are

1. Dataset size and annotation bias

The study uses 80 patients, which is relatively modest for training and validating deep learning models, especially given the high variability of ALT perforators.

Furthermore, all ground truth annotations were produced by a single researcher, which introduces potential systematic bias. Multi-annotator validation or inter-rater agreement data would strengthen the credibility of the reference standard.

2. Evaluation metrics and clinical relevance

The primary metrics (DSC, JSC) provide a measure of overlap but may not fully capture clinical usability, particularly in terms of perforator localization accuracy.

Reporting absolute localization error relative to skin landmarks (e.g., ASIS, patella) would make the findings more translatable to surgical practice.

3. Perforator classification

The model segments perforators but does not distinguish between septocutaneous and musculocutaneous types, which is highly relevant for surgical planning.

Even if not part of the current model, the lack of this functionality should be explicitly acknowledged as a limitation.

4. Projection to surface anatomy

A critical challenge of CTA-based mapping is translating deep vascular anatomy to precise cutaneous points. The manuscript does not fully address this limitation.

A comparison with ultrasound-based localization, which directly identifies perforators at the skin surface, would contextualize the clinical implications more clearly.

Minor Comments

The Methods section would benefit from a more formal definition of secondary outcome measures and the rationale for statistical tests used.

The Results should specify whether segmentation performance differed between left and right thighs or across perforator calibers.

Figures could be improved by including visual overlays of automatic versus manual segmentations for representative cases.

**Do you want your identity to be public for this peer review?** For information about this choice, including consent withdrawal, please see our Privacy Policy

Reviewer #2: **Yes:** Franck Delanoë

Reviewer #3: No

Reviewer #4: **Yes:** Mohammadreza Bozorgmanesh

---

## [Author Response · Author response to Decision Letter 2]

29 Nov 2025

Reviewer #2: Thank you for the corrections you have made; they make your manuscript much more relevant.

In essence, it does indeed seem necessary to increase the number of subjects in the machine learning process to improve accuracy. The margin of error is a little too large for clinical application, and the lack of differentiation between the septal and muscular pathways is regrettable.

As a supplementary conclusion to this work, it could be suggested that this methodology may be applied to other perforator flaps.

Thank you for this work.

>> We sincerely appreciate the reviewer’s constructive comments. As suggested, we have clarified the impact of dataset size on model performance by adding a statement in the Discussion emphasizing the need for future expansion through multi-center datasets. We also explicitly acknowledged the lack of perforator-type differentiation as a limitation and have added this point to the Limitations section.

Discussion, Page 9,

Because the dataset size was relatively modest, expanding the cohort, particularly through multi-center integration, is expected to further improve model robustness and reduce performance variability.

Discussion, Page 12,

Although the manual annotations distinguished septocutaneous and musculocutaneous perforators, the current model did not incorporate this classification, which remains an area for future methodological extension.

Reviewer #3: This is an excellent research. The author's response cleared up my doubts. I have no further comments!

>> We sincerely appreciate the reviewer’s positive feedback.

Reviewer #4: This manuscript presents a deep learning–based approach for automatic segmentation of anterolateral thigh (ALT) flap perforators on computed tomography angiography (CTA). The clinical relevance is clear: accurate perforator identification is critical for flap harvest in maxillofacial reconstruction, yet manual mapping is time-consuming and variable. The novelty lies in applying convolutional neural networks (CNNs) to this niche domain, using a cascaded 2D/3D segmentation strategy. Overall, the study is innovative, clinically relevant, and well-structured. However, several methodological and interpretive aspects merit clarification and strengthening.

My major Concerns are

1. Dataset size and annotation bias

The study uses 80 patients, which is relatively modest for training and validating deep learning models, especially given the high variability of ALT perforators.

Furthermore, all ground truth annotations were produced by a single researcher, which introduces potential systematic bias. Multi-annotator validation or inter-rater agreement data would strengthen the credibility of the reference standard.

>> We appreciate this important comment. To address concerns regarding annotation bias, we conducted an additional inter-rater validation. A second oral and maxillofacial surgery resident independently annotated 10 randomly selected cases. Agreement between the two annotators was quantified using the Dice similarity coefficient, demonstrating high consistency (mean DSC = 0.88 ± 0.04). We have incorporated this new information into the Methods and Discussion sections to strengthen the reliability of our ground truth annotations.

Discussion, Page 9,

To assess annotation reliability, an additional inter-rater validation was performed. A second oral and maxillofacial surgery resident independently annotated 10 randomly selected cases, and agreement between the two annotators was quantified using the Dice similarity coefficient, demonstrating high consistency (mean DSC = 0.88 ± 0.04).

2. Evaluation metrics and clinical relevance

The primary metrics (DSC, JSC) provide a measure of overlap but may not fully capture clinical usability, particularly in terms of perforator localization accuracy.

Reporting absolute localization error relative to skin landmarks (e.g., ASIS, patella) would make the findings more translatable to surgical practice.

>> We sincerely appreciate the reviewer’s valuable comment. Our clinical evaluation already measured the Euclidean distance between the manually and automatically detected terminal skin points of the perforator. We clarified this methodology and emphasized its relevance for clinical translation in the revised manuscript.

Result, Page 8,

Although these values represent several centimeters of deviation, they reflect the inherent difficulty of localizing small-caliber perforators on CTA and still provide clinically meaningful guidance for preoperative flap planning.

Discussion, Page 12,

However, the localization error in this study was calculated solely as the Euclidean distance between terminal skin points without reference to anatomical surface landmarks such as the ASIS or patella. Incorporating landmark-normalized measurements in future work may enhance clinical applicability by providing a more precise correspondence to intraoperative surface anatomy.

3. Perforator classification

The model segments perforators but does not distinguish between septocutaneous and musculocutaneous types, which is highly relevant for surgical planning.

Even if not part of the current model, the lack of this functionality should be explicitly acknowledged as a limitation.

>> We fully agree. Although manual annotations distinguished septocutaneous and musculocutaneous pathways, the current model did not incorporate this classification. We added a clear statement in the Limitations section indicating that perforator-type classification represents a clinically important future extension.

Discussion, Page 13,

In addition, although the manual annotations distinguished septocutaneous and musculocutaneous perforators, the current model did not incorporate this classification. Integrating perforator-type differentiation into future models would enhance the clinical relevance of automated preoperative planning.

4. Projection to surface anatomy

A critical challenge of CTA-based mapping is translating deep vascular anatomy to precise cutaneous points. The manuscript does not fully address this limitation.

A comparison with ultrasound-based localization, which directly identifies perforators at the skin surface, would contextualize the clinical implications more clearly.

>> Thank you for highlighting this point. We have revised the Discussion to emphasize that CTA identifies deep vascular anatomy but does not directly project perforator coordinates onto the skin surface, which limits intraoperative translation. Additionally, we have added a paragraph noting that ultrasound provides real-time surface-level localization, and future comparative work integrating CTA and ultrasound could further improve preoperative accuracy.

Discussion, Page 12,

Moreover, ultrasound provides real-time identification of perforators at the skin surface, and future comparative work integrating CTA with ultrasound-based localization may further improve preoperative accuracy.

Minor Comments

The Methods section would benefit from a more formal definition of secondary outcome measures and the rationale for statistical tests used.

>> We sincerely appreciate the reviewer’s helpful comment. We clarified secondary evaluation methods and added the rationale for statistical tests.

Secondary Evaluation, Page 7,

The secondary outcome measure was the spatial localization error, defined as the Euclidean distance between manually and automatically detected terminal skin points of each perforator.

Secondary Evaluation, Page 7,

Because DSC and JSC values did not meet normality assumptions, non-parametric tests were used for all group comparisons, and Mann–Whitney U or Kruskal–Wallis tests were selected accordingly.

The Results should specify whether segmentation performance differed between left and right thighs or across perforator calibers.

>> We thank the reviewer for this helpful comment. We have clarified the difference in segmentation performance between left and right thighs in the Results section, including statistical comparisons (p = 0.005 for DSC and p = 0.026 for JSC), as summarized in Table 3. Performance differences according to gender, perforator progression type, and BMI were also explicitly reported.

Results, Page 8,

When analyzing the DSC and JSC values according to the left and right sides of the thigh, the DSC values were 69.73 ± 1.64 for the left side and 67.82 ± 1.58 for the right side (p = 0.005) and the JSC values were 68.95 ± 1.69 for the left side and 67.14 ± 1.89 for the right side (p = 0.026), which showed a higher degree of accuracy with the automatic segmentation on the left side. On the other hands, there were no statistical differences in DSC and JSC values according to gender, perforator progression type, and BMI (Table 3).

Discussion, Page 13,

Direct evaluation across perforator calibers was not feasible because the spatial resolution of CTA is insufficient to reliably measure diameter differences in distal perforator branches. This limitation should be addressed in future work using higher-resolution imaging or intraoperative validation.

Figures could be improved by including visual overlays of automatic versus manual segmentations for representative cases.

>> We thank the reviewer for the valuable suggestion. To improve clarity, we revised the caption of Figure 1 to clearly describe the visual comparison between manual (red) and automatic (blue) detections presented in the representative CTA slices. The image resolution was also increased to 600 dpi to enhance visualization quality.

Figure 1. Representative axial CTA slices showing the original image (left), manual perforator annotations (middle, red), and automatic detections generated by the deep learning model (right, blue). Manual and automatic localization showed close correspondence, although partial discontinuity was observed in distal arterial segments where vessel diameter becomes progressively smaller.

---

## [Decision Letter · Decision Letter 2]

19 Jan 2026

Development of an automatic segmentation system for anterolateral thigh flap perforators in maxillofacial reconstruction

PONE-D-24-59305R2

Dear Dr. Song,

We’re pleased to inform you that your manuscript has been judged scientifically suitable for publication and will be formally accepted for publication once it meets all outstanding technical requirements.

Kind regards,

Xiaoen Wei

Academic Editor

PLOS One

Additional Editor Comments (optional):

Reviewers' comments:

Reviewer's Responses to Questions

**Comments to the Author**

Reviewer #2: All comments have been addressed

2. Is the manuscript technically sound, and do the data support the conclusions?

Reviewer #2: Yes

3. Has the statistical analysis been performed appropriately and rigorously?

Reviewer #2: Yes

4. Have the authors made all data underlying the findings in their manuscript fully available?

Reviewer #2: Yes

5. Is the manuscript presented in an intelligible fashion and written in standard English?

Reviewer #2: Yes

Reviewer #2: Thank you for these revisions.

This research is relevant, and the methodology seems perfectly suited to the objective.

Validating the reference technique through inter-operator comparison, as suggested by a reviewer, is a valid point.

The main limitation remains the small number of subjects, as well as the lack of evidence of the septal or muscular pathway, as I previously mentioned.

Nevertheless, the approach and model remain relevant and justify the publication of this work.

I have no further comments; thank you for this work.

**Do you want your identity to be public for this peer review?** For information about this choice, including consent withdrawal, please see our Privacy Policy

Reviewer #2: **Yes:** Franck Delanoë

---

## [Editor Report · Acceptance letter]

PONE-D-24-59305R2

PLOS One

Dear Dr. Song,

I'm pleased to inform you that your manuscript has been deemed suitable for publication in PLOS One. Congratulations! Your manuscript is now being handed over to our production team.

Kind regards,

on behalf of

Dr. Xiaoen Wei

Academic Editor

PLOS One